# Fouling Mitigation by Optimizing Flow Rate and Pulsed Electric Field during Bipolar Membrane Electroacidification of Caseinate Solution

**DOI:** 10.3390/membranes11070534

**Published:** 2021-07-15

**Authors:** Vladlen S. Nichka, Victor V. Nikonenko, Laurent Bazinet

**Affiliations:** 1Department of Food Sciences, Laboratoire de Transformation Alimentaire et Procédés ÉlectroMembranaires (LTAPEM, Laboratory of Food Processing and Electromembrane Processes), Dairy Research Center (STELA), Institute of Nutrition and Functional Foods (INAF), Université Laval, Québec, QC G1V 0A6, Canada; vladlen.nichka.1@ulaval.ca; 2Membrane Institute, Kuban State University, 149 Stavropolskaya Str., 350040 Krasnodar, Russia; v_nikonenko@mail.ru

**Keywords:** electrochemical acidification, electrodialysis, casein, concentration polarization, ion-exchange membrane, fouling, Reynolds number, mode of current, pause duration

## Abstract

The efficiency of separation processes using ion exchange membranes (IEMs), especially in the food industry, is significantly limited by the fouling phenomenon, which is the process of the attachment and growth of certain species on the surface and inside the membrane. Pulsed electric field (PEF) mode, which consists in the application of constant current density pulses during a fixed time (Ton) alternated with pause lapses (Toff), has a positive antifouling impact. The aim of this study was to investigate the combined effect of three different relatively high flow rates of feed solution (corresponding to Reynolds numbers of 187, 374 and 560) and various pulse–pause ratios of PEF current regime on protein fouling kinetics during electrodialysis with bipolar membranes (EDBM) of a model caseinate solution. Four different pulse/pause regimes (with Ton/Toff ratios equal to 10 s/10 s, 10 s/20 s, 10 s/33 s and 10 s/50 s) during electrodialysis (ED) treatment were evaluated at a current density of 5 mA/cm^2^. It was found that increasing the pause duration and caseinate solution flow rate had a positive impact on the minimization of protein fouling occurring on the cationic surface of the bipolar membrane (BPM) during the EDBM. Both a long pause and high flow rate contribute to a more effective decrease in the concentration of protons and caseinate anions at the BPM surface: a very good membrane performance was achieved with 50 s of pause duration of PEF and a flow rate corresponding to Re = 374. A further increase in PEF pause duration (above 50 s) or flow rate (above Re = 374) did not lead to a significant decrease in the amount of fouling.

## 1. Introduction

Electrodialysis (ED) is a process in which electrically charged membranes are used to separate ions from an aqueous solution under the effect of an electric potential difference. One of the most interesting electrodialytic processes for the food industry is electrodialysis with bipolar membranes (EDBM). The main advantage of EDBM is the absence of chemical additions for adjusting the pH of the solution under treatment, which makes it particularly attractive in the food and pharmaceutical sectors where decreasing environmental pollution is one of the main actual concerns. It has a wide range of potential applications in such processes as the deacidification of fruit juices [1], the demineralization of whey proteins [2], the separation of organic acids produced by fermentation [3], etc. The production of casein, the major milk protein, by means of EDBM is a very interesting method in the sustainable development context due to the high product purity, and the absence of waste generation and hazardous reagents [4].

Among the most important problems during EDBM is the fouling of ion exchange membranes (IEM). The fouling of membranes degrades their performance by increasing their diffusion permeability [5], and decreasing their permselectivity [6], electrical conductivity and exchange capacity [7] as well as increasing the energy costs of the whole ED process [8]. All these factors can lead to a regular replacement of membranes, which significantly increases the cost of ED [9]. Indeed, at an industrial scale, the cost of the IEMs and spacers contributes to 25–30% of the total cost of a fully automated ED: in addition, this percentage increases when the level of automation is decreased [10].

One of the ways to prevent and minimize fouling is the use of pulsed electric fields (PEFs). The PEF mode is a non-stationary regime of supplying electric current for a certain time Ton (pulse lapse), followed by a pause lapse with a duration of Toff, when the current is turned off [10]. The use of PEF in ED was first proposed to control membrane ion-transport selectivity in order to separate Na^+^ and Ca^2+^ [11]. Among the main advantages of PEF is the suppression of the concentration polarization phenomenon, which is the formation of concentration profiles at the membrane surface due to the difference between the transport numbers in the membrane and solution, and consequently a decrease in water dissociation [12]; an increase in the current efficiency, and consequently a decrease in the relative energy consumption (EC) [13]; an increase in the effectiveness against different types of fouling caused by humate [14], peptides/amino acids [15] and lignin [16], etc. Moreover, ED with PEF has been used to reduce the fouling rate of IEM containing organic molecules [17] and as an effective antifouling method for the ED of fermentation waste [18]. In 2007, Ruiz and coauthors [19] were the first to demonstrate the efficiency of using PEFs to eliminate protein fouling from the AEM diluate side during ED of casein solution. Recently, in the work of Nichka and coauthors [20], an analysis of combined effect of solution flow rate and PEF or continuous current (CC) regime on protein fouling kinetics on the cationic layer of bipolar membrane (BPM) during EDBM of skim milk was carried out. In that work, the effect of five flow rates on the fouling kinetics was investigated, but only in one PEF condition with the Ton/Toff equal to 10 s/50 s. It has been found that the application of a PEF current regime, as well as an increase in the flow rate to a sufficiently high value (corresponding to Re = 485) makes it possible to almost completely eliminate protein fouling on the membrane surface.

In this context, the aim of the present study was to investigate the influence of PEF parameters in combination with different hydrodynamic conditions on BPM protein fouling during EDBM when using a model caseinate solution. The caseinate solution was chosen instead of skim milk (which is usually treated in the food industry) in order to more clearly define the conditions under which protein fouling occurs (or does not occur)—taking into account the fact that the composition of caseinate solution is simpler than that of milk and allows for isolating protein fouling with no interference of divalent ion scaling. The main objectives of the research were to study: (1) the effect of PEF ratios on protein-specific fouling formation or removal; (2) the impact of flow rate conditions on fouling kinetics; (3) the combined effect of these two factors and, finally; (4) to determine the optimal antifouling strategy. The novelty of the work is that the coupled impact of PEF regime in combination with increased flow rates of solution during EDBM of caseinate solution, as far as we know, has never been investigated before.

## 2. Materials and Methods

### 2.1. Materials

Casein sodium salt (or sodium caseinate) used in this study was purchased from Sigma Chemical Co. (St. Louis, MO, USA). Na_2_SO_4_ (ACS grade) and KCl (ACS grade) were obtained from BDH (VWR International Inc., Mississauga, ON, Canada). Chemicals for cleaning-in-place of the ED system (HCl and NaOH), were purchased from Fisher Scientific (Montreal, QC, Canada).

The caseinate powder was reconstituted with distilled water in order to have the same content of casein as in a skim milk, which was described in a previous study [20]. Thus, the total protein content in the model caseinate solution is 27 g/L. Solution was held overnight under gentle agitation at 10 °C to allow its complete solubilization and rehydration. Moreover, potassium chloride salt was added in the model solution before each experiment in order to have the same initial conductivity as for skim milk (about 3100 µs/cm) [20].

The average mineral composition of raw caseinate solution is presented in Table 1. The data was obtained by ICP-OES (Inductively Coupled Plasma-Optical Emission Spectrometry) analysis.

Casein sodium salt is made from casein of bovine milk, the main protein presented in milk. Casein, which makes up approximately 80% of the total protein in bovine milk is presented in four forms: α-s1 Casein, α-s2 Casein, β-Casein, and κ-Casein. The approximate casein composition of milk is (g/L): α-s1 (12–15); α-s2 (3–4); β (9–11); and κ (2–4). The casein subunits vary primarily in molecular weight, isoelectric point, and level of phosphorylation. Table 2 lists these differences [21,22].

### 2.2. Methods

#### 2.2.1. Electrodialysis Cell

A microflow-type cell (Electro-Cell AB, Täby, Sweden) was used for the experiments (Figure 1), consisting of four compartments with one Neosepta cationic membrane (CMX-fg) and two Neosepta bipolar membranes (BP-1E) (Astom, Tokyo, Japan). The membranes used in the study had an equal effective surface area of 10 cm^2^. The anode was a plate dimensionally stable electrode (DSA-O2, Ti/IrO_2_ coating) and the cathode was a 316-stainless-steel electrode. The tested system defined three closed loops containing equal volumes of 300 mL of 20 g/L Na_2_SO_4_ electrolyte solution, 2 g/L KCl aqueous solution and the model casein solution. The flow rates were equal for each solution used in the experiment. Each closed loop was connected to an external plastic tank, allowing a continuous recirculation of solutions. The variation in temperature during the whole process was about 12 °C (from 25 °C to 37 °C). This variation was the same for each mode of current. In the current study, the ED system was not equipped to maintain a constant temperature. Controlling the temperature during ED process would use lots of energy, and the possible energy saving related to the fouling mitigation would be lost. Thus, to be closer to the reality and to test all experimental parameters in real conditions, the temperature increase was consequently not controlled.

#### 2.2.2. Protocol

EDBM was carried out under a constant current intensity of 50 mA, corresponding to a current density of 5 mA/cm^2^ generated by a Xantrex power supply (Model HPD 60-5SX; Burnaby, BC, Canada). The active membrane surface was 10 cm^2^. Experiments were conducted under the PEF current regime with different pulse–pause ratios (10 s–10 s, 10 s–20 s, 10 s–33 s, 10 s–50 s). In addition, one complementary experiment with 10 s–100 s PEF ratio was carried out to test whether a significant increase in the PEF pause time would affect the fouling kinetics. The current density (5 mA/cm^2^) was the same during the pulse lapses (10 s in all cases) in PEF mode regardless of the pause duration. Three different linear velocities (7.8, 15.6 and 23.4 cm/s, respectively) were applied for each PEF current regime to test the combined effect of both factors: PEF current modes and hydrodynamic conditions on protein fouling formation during EDBM of model caseinate solution. Inside the channels of the ED cell, there were spacers for flow turbulization (Figure 2) with the length of 6 cm, width of 3 cm and filament diameter of 0.07 cm.

Three replicates of each combination (flow rate–PEF regime) were performed in this experiment for a total of 36 experiments. PEF was generated by a modified Pulsewave 760 generator from Bio-Rad laboratories (Richmond, BC, Canada). The duration of experiments with different PEF ratios was equivalent to 30 min of conventional ED with the same constant current density (5 mA/cm^2^) in order to compare the different PEF regimes with respect to the same amount of charges (about 90 C). Samples of concentrate solution (KCl) with volume of 10 mL were collected before and after each ED treatment to investigate the ion migration process. These samples were frozen at −30 °C after experiments until the determination of calcium, potassium, sodium and phosphorus concentrations, performed by ICP-OES (Inductively Coupled Plasma–Optical Emission Spectrometry), and chlorine concentration, determined by flow injection analysis.

#### 2.2.3. Analyses and Calculations

It is worth mentioning that the determination of the pH, as well as conductivity of caseinate and KCl solutions, was conducted in the corresponding external intermediate tanks shown in Figure 1.

##### Solution Conductivity

The conductivity of model caseinate and KCl solutions in the streams passing through the cell compartments was measured with an YSI conductivity meter Model 3100 with an YSI immersion probe Model 3252, cell constant K = 0.1 cm^−1^ (Yellow Springs Instrument Co., Yellowsprings, OH, USA).

##### Solution pH

pH-meter Model SP70P from VWR International (Montreal, QC, Canada) was used to measure the pH of caseinate stream. The pH of KCl stream was measured with an Orion Star A221 pH-meter from Thermo Scientific (Montreal, QC, Canada).

##### Membrane Thickness and Electrical Conductivity

All membranes tested were soaked in a 0.5 M NaCl solution for 30 min before and after each experiment for their thickness and conductivity characterization. The membrane thicknesses were measured using an electronic digital micrometer from Marathon watch company LTD (Richmond Hill, ON, Canada). The micrometer had a 10-mm-diameter flat contact point. Each membrane thickness value was measured at six different locations on the effective surface of membrane and then averaged [23]. The electrical conductivity of membrane was calculated from the values of measured thicknesses and electrical resistances, obtained from the membrane conductance. The conductance was measured with an YSI conductivity meter Model 3100 (Yellow Springs, OH, USA) equipped with a specially designed clip as described in [23]. A 0.5 M NaCl solution was used as a reference solution. Six conductance measurements of the reference solution and of the membrane in the reference solution were taken. The membrane conductance in the reference solution was taken on the effective membrane surface. As described by Lteif et al. [24] and Lebrun et al. [25], the membrane electrical resistance was then calculated according to Equation (1):(1)$$R_{m}=\frac{1}{G_{m}}=\frac{1}{G_{m+S}}-\frac{1}{G_{S}}=R_{m+s}-R_{s},$$
where *R_m_* and *G_m_* are the transverse electric resistance (in Ω) and conductance (in S) of the membrane, respectively; *R_m+s_* and *G_m+s_* are the resistance (in Ω) and conductance (in S) of the membrane and the reference solution; *R_s_* and *G_s_* are the resistance (in Ω) and conductance (in S) of the reference solution.

Membrane conductivity was calculated from Equation (2) [15]:(2)$$\kappa =\frac{L}{R_{m}A},$$
where *κ* is the membrane electrical conductivity (in S/cm), *L* the thickness of the membrane (in cm) and *A* the electrode area (1 cm^2^).

##### Foulant Amount

Protein fouling was collected from the membrane surfaces right after ED cell dismounting, and then freeze-dried in a Labconco lyophyliser, Model Freezone 4.5 (Kansas City, MO, USA). After weighing, the dried powder samples were kept at −20 °C until further analysis.

##### Membrane Surface Photographs

Digital photographs of the cationic interfaces of BPMs 1, which was in contact with the model caseinate solution, were taken after each EDBM experiment for graphical representation of foulant deposition.

##### Number of Charges Transported

Equation (3) was used for the number of charges transported calculation:(3)Q=It,
where *Q* is the number of charges transported (in C), *I* is the current intensity (in A), *t* is the duration (in s) of the electric current flow excluding the pause lapses. *I* was constant during the pulse period and zero during the pauses.

##### Energy Consumptions

The EC (in Wh) was calculated from Equation (4):(4)EC=I∫U(t)dt,
where *I* is the current intensity (in A), *U(t)* the voltage (in V) recording during experiments as a function of time [19]. The effective time of current application taken into account was the time during the pulse periods.

##### Reynolds Numbers

Usually, the Reynolds number for membrane stacks is defined as [26]:(5)Re=ρνhμ,
where ρ is the density of the fluid (in kg/m^3^), ν is the average flow velocity (in m/s), *h* is the intermembrane distance (in m), *µ* is the dynamic viscosity of the fluid (in Pa s).

The values of density ρ and dynamic viscosity *µ* for the calculation of Reynolds number were taken as for skim milk, which are close to the characteristics of caseinate solution. It is known that density and viscosity are temperature-dependent parameters; therefore, the Reynolds number is also dependent on temperature. Under the conditions of our experiment, the density values ρ varied in the range from 1031 to 1027 kg/m^3^, viscosity *µ*, from 0.0015 to 0.0011 Pa s [27].

##### Statistical Analyses

All data obtained in the experiments were subjected to a two- or three-way analysis of variance (ANOVA) using SigmaPlot software (SigmaPlot version 12.5, Chicago, IL, USA). Holm–Sidak tests (significance level of *p =* 0.05) was used to define the effect of each factor considered and their combination on protein fouling kinetics.

## 3. Results and Discussion

### 3.1. Evolution of pH

As mentioned above, the duration of all studied EDBM processes was determined by a fixed amount of electricity passed through the ED cell (Section 2.2.2); this amount (90 C) corresponds to a 30-min treatment of the caseinate solution in the conventional ED mode with a current density of 5 mA/cm^2^.

As mentioned above (Section 2.2.3), the Reynolds number is a temperature dependent parameter, so it varied during the EDBM process when the linear velocity was kept constant. Under the described experimental conditions, the Reynolds numbers were in the range of 187 ± 30, 374 ± 60 and 560 ± 90, corresponding to the linear velocities of 7.8, 15.6 and 23.4 cm/s, respectively. In this paper the average values of Reynolds numbers will be used as indicative parameters for the description of fouling kinetics. It should be noted that under all the considered hydrodynamic conditions, a turbulent fluid flow occurred in the ED channels, since the Reynolds numbers are higher than the critical Re value for channels with similar separators, which is close to 100 [28].

#### 3.1.1. Caseinate Stream

The analysis of variance showed that the PEF regime (*p* < 0.001) and flow rate (*p* = 0.024) have a significant impact on the variation in pH of caseinate stream during EDBM, while the coupled effect of flow rate/PEF regime (*p* > 0.05) has no effect. A slow linear decrease in pH with increasing time of solution treatment was observed for all the experimental conditions considered. In all cases, the maximum and minimum pH values were 6.5 and 6.2, respectively. The decrease in pH can be explained by the addition of hydrogen ions from the cation-exchange layer of BPM 1 due to water splitting at the bipolar junction (Figure 1). However, not all the H^+^ ions generated by the BPM contribute to the pH decrease; some of them became bounded. There are two possible mechanisms for bounding the H^+^ ions. Some of the H^+^ ions can interact with negatively charged casein ions with the formation of an uncharged form that precipitates [29]:(6)$$H^{+}+{\mathrm{Cas}}^{-}={\mathrm{CasH}}^{0},$$

The characteristic / threshold pH value for the last reaction is around 4.6.

Additionally, hydrogen ions could interact with phosphates of caseinate. However, the only phosphorus present in the caseinate solution is in the phosphoserine or phosphoseryl residues of the different caseins (Table 2) in the form of H_2_PO_3_ (H-Ser(PO3H2)-OH with a pKa = 5.6). The analysis showed that the concentration of H_2_PO_3_ is five times lower than the concentration of casein (0.0026 mol/L vs. 0.013 mol/L, respectively) in the model solution. Therefore, it does not interfere with or has a really low impact on H^+^ bounding, so this mechanism can be neglected.

Therefore, the relatively slow rate of caseinate acidification is linked to the buffer capacity of the caseinate solution [30], for which proteins are the responsible species (Equation (6)). As Figure 3 shows (and according to the statistical analysis), the difference in pH between the final (obtained after passing 90 C of electricity) and initial solution pH increases with increasing the pause duration (gradually from 10 s to 50 s). This rule holds true for all considered flow rates, but the difference in pH also increases with an increase in the flow rate. Probably, in the case of long pauses of PEF and high flow rates of the solution, the concentration of H^+^ ions at the BPM 1 surface is too low for protein deposit formation according to Equation (6). Therefore, a significant number of the H^+^ ions generated by BPM 1 remain in an unbound state. Thus, their concentration in the external tank of the desalination stream, where the pH of caseinate is measured, is relatively high. On the other hand, in the case of short pauses and low flow rates, the local concentration of H^+^ ions at the BPM 1 surface is higher and exceeds the critical value required for the formation of protein fouling (pH = 4.6). Then, most of these ions are bounded forming non-charged proteins. As a result, the H^+^ ion concentration in the unbound state in the caseinate stream is relatively low.

#### 3.1.2. KCl Stream

It appeared from the statistical analysis that the PEF regime (*p* = 0.074) and flow rate (*p* = 0.885) as well as the coupled effect of these factors (*p* > 0.05) had no significant effect on the variation in pH of the KCl stream during EDBM (Figure 4). Indeed, whatever the combination or conditions tested, the pH increased from 6.2 to approximately 10; this increase is due to the generation of the OH^–^ ions from the BPM 2 (Figure 1) into the concentrate compartment. A fast increase in pH appeared at the beginning of the process (up to 20 C of transferred charge) and then the pH value remained almost constant until the end of each experiment: such a shape is due to the fact that pH is a logarithmic value of the H^+^ concentration. However, the variation in pH between the beginning of the process and its end in the KCl stream is higher than in the caseinate stream (4.2 vs. 0.3, respectively) due to the difference in their buffer capacities: the caseinate solution has a high buffer capacity, as discussed in Section 3.1.1, whereas the KCl solution has zero buffer capacity. The stabilization of the pH value in the KCl stream could be caused by increasing the transport of the H^+^ ions from the caseinate stream through the CEM over time as well as by the leakage of the OH^–^ ions through this membrane.

### 3.2. Cake Layer Foulant Morphology and Density

No fouling was observed on both interfaces of the CEM and BPM 2, as well as on the anionic layer of the BPM 1 (data not shown), regardless of the hydrodynamic and PEF conditions applied. However, digital photographs of the surface of the BPM 1 cationic layer, which was in contact with the caseinate stream, show the presence of protein fouling for all the conditions applied (Figure 5).

The reaction leading to the protein precipitation is described by Equation (6), which shows that when H^+^ ions interact with negatively charged casein ions, an uncharged protein (the casein) is formed. Therefore, the high concentrations of both hydrogen ions and caseinate anions at the BPM 1 surface promote the formation of neutral casein leading to a deposit on the cation-exchange layer surface. The H^+^ ions are generated at the membrane bipolar junction; the caseinate anions are transported from the bulk solution to the cation-exchange layer surface under the action of the external electric field. It is worth mentioning that the critical pH value for this reaction is 4.6. When there is an excess in the concentration of both ions, reaction (Equation (6)) occurs from the left to the right.

In the external tank, where pH was measured, such a value of pH was not achieved in the experiments (Figure 3), but it is possible that it is reached locally at the BPM 1 surface. The results of the experiments suggest that this critical value is achieved in the cases where the pause duration is short and the flow rate is low.

It can be seen from Figure 5 that at short pauses of PEF, a transparent gel-like protein deposit, predominates, while at longer pauses, the fouling is mainly white. In this regard, visually it may seem that in the case of short pauses of PEF, less protein fouling is observed. The color of the deposit can be connected with a thermodenaturation process of protein during the EDBM process [31]. Apparently, in the cases of long PEF pauses and elevated flow rates, the temperature of solution is lower due to a more active mixing and more effective heat dissipation. Thus, the temperature for the protein denaturation process was not achieved. During short PEF pauses, a higher temperature was reached, which led to thermodenaturation of the protein with the formation of a gel-like deposit. Protein precipitation in the form of a gel was also observed by Ruiz and coauthors [19] during ED of a caseinate solution.

For short pause durations of PEF (10 s), even the separator between the membranes was contaminated by fouling. For pauses of 20 s, the major part of the fouling was concentrated on the membrane surface, not inside the separator. A pause duration of 33 s significantly decreased the amount of protein fouling on the membrane surface, while a pause duration of 50 s allowed avoiding the fouling formed on the BPM almost completely, especially at high flow rates (Figure 6). The positive coupled effect of pause duration of PEF and flow rate on the protein fouling formation can be explained by the fact that the excess of hydrogen ions and caseinate anions at the BPM 1 cation-exchange layer surface dissipates due to two factors: a long pause (1) and a high flow rate (2). In addition, a high-speed solution flow can flush away the previously accumulated protein deposits from the membrane surface. The obtained results confirm the earlier findings, which showed that in the case of protein solutions, long PEF pauses during EDBM promote the removal of macromolecule fouling from the IEM surface [19,32].

ANOVA showed that there is an effect of PEF regime (*p* < 0.001) and flow rate (*p* < 0.001) as well as coupled effect PEF regime/flow rate (*p* = 0.029) on the weight of protein fouling recovered on each BPM cationic interface after EDBM treatment. It was clearly shown that increasing the pause duration of PEF as well as the flow rate of the solution leads to a decrease in protein fouling formation (−87% in weight of protein fouling). Besides, one complementary experiment with 10 s−100 s PEF ratio was carried out to check the possibility of the complete removal of fouling from the membrane surface and it appeared that the weight of protein fouling recovered was in a range of standard deviation (SD) of 10 s–50 s values, and allowed us to conclude that there is no significant impact of increasing the pause duration of PEF over 50 s for fouling minimization: 50 s being quite an optimal condition with a flow rate which corresponds to Re = 374. It should be also mentioned that there is no impact of flow rate on the amount of fouling for the 10 s–10 s PEF condition. For 10 s–20 s, 10 s –33 s and 10 s–50 s PEF conditions, the main impact of flow rate was observed until the flow rate reached a Re of 374; a further increase in flow rate did not lead to significant changes in the amount of protein fouling recovered. The decrease in the amount of protein fouling on a BPM 1 surface with an increase in solution flow rate was observed due to the creation of adverse hydrodynamic conditions for the attachment and growth of protein fouling. It is a confirmed fact that increasing the flow rate is effective for fouling mitigation [33,34]. The positive effect of the flow rate and the use of PEF for fouling mitigation also consists in reducing concentration polarization [35]. A lower concentration allows avoiding the accumulation and close packing of foulants at the membrane surface combined with the cleaning effect of the solution recirculation, which was mentioned earlier [10].

### 3.3. Evolution of Conductivity

#### 3.3.1. Caseinate Stream

ANOVA showed that the PEF regime (*p* < 0.001) has a significant effect on the variation in caseinate conductivity during EDBM while the flow rate (*p* > 0.05) and the coupled effect of flow rate/current regime (*p* > 0.05) have no effect. Figure 7 shows the difference between the conductivity of the caseinate solution after transferring a certain amount of charges in an EDBM process and the conductivity of the initial solution. The variation in conductivity of caseinate stream was influenced by the desalination and acidification phenomena in the stream. The cations of model caseinate solution migrate through the CEM into the KCl stream, contributing to decrease the caseinate stream conductivity; at the same time, an opposite process of conductivity increase takes place due to the hydrogen ions generated by the BPM 1 (Figure 1). However, the phenomenon of acidification by the generation of H^+^ ions contributes more to the conductivity variation by increasing the caseinate conductivity because of the higher mobility of hydrogen ions [30]. Moreover, the higher the concentration of the H^+^ ions, the higher the conductivity of the caseinate solution. As shown above (Figure 3), the concentration of the H^+^ ions in this solution increases (the pH decreases) with increasing the pause duration. Consequently, the caseinate stream conductivity, as expected, is the highest in the case of the longest pause duration. The lowest conductivity was found in the case of the shortest pause durations (Figure 7).

It should be also noted that in the case of long pauses, the system comes to a stationary state faster (Figure 7c,d). In the case of 50 s pause lapse, the stationary state was reached earlier (at around 50 C or 17 min of time equivalent to conventional ED). For 20 s and 10 s pause durations, the steady state was not reached until the end of the experiments. The time required for the achievement of a steady-state plateau of the conductivity is apparently connected with the kinetics of fouling on the BPM. In the case of short pause durations, the amount of protein deposits on BPM 1 is significant (Section 3.2). This process takes time, and protons are involved in it. During the deposit formation, the H^+^ ions are consumed, which does not allow the steady state to be quickly reached. When the pause duration is long, the amount of deposit is quite small, and a steady state is rapidly established.

#### 3.3.2. KCl Stream

It appeared from the statistical analysis, that the PEF regime and flow rate as well as the coupled effect of these factors had no significant effect (*p* > 0.05) on the variation in KCl conductivity (Figure 8). For all the cases considered, there was a slow increase in conductivity of about 900 µS/cm (which is 28% of the initial conductivity) until the end of the experiment. For the KCl stream both processes—electromigration of cations through the CEM and generation of hydroxide ions by BPM 2 lead to an increase in KCl conductivity during the EDBM process. The same effects of the OH^−^ ions’ delivery into the KCl solution and the ion migration from the caseinate solutions on the conductivity were described by Ruiz et al. [19]. It should be also mentioned that the hydroxyls will have a higher impact on the conductivity value since their mobility is higher in comparison with the salt ions [36].

### 3.4. Mineral Content of KCl Stream

Table 3 shows the mineral composition of KCl stream samples before and after EDBM treatment as well as the variation in concentrations between the final and initial values. The concentrations of Ca^2+^, K^+^, Na^+^, P were determined by ICP-OES (Inductively Coupled Plasma–Optical Emission Spectrometry) analysis; concentration of Cl^−^ was obtained by flow injection analysis. Concerning Na^+^ concentration, ANOVA analysis showed that there is a significant impact of both factors considered–PEF regime (*p* < 0.001) and flow rate (*p* < 0.001) on its concentration variation. It could be seen from Table 3 that there is an increase in Na^+^ concentration after ED treatment. This is connected with the fact that during ED there is a migration of Na^+^ from the desalination to the concentration chamber through the CEM. It should be mentioned that sodium has the highest concentration among the other ions (K^+^ and Ca^2+^) presented in the caseinate stream. The concentrations of these cations are very low, therefore they could not significantly interfere with its migration or to contribute to the global cation migration (Table 1). In addition, there is an increase in the final Na^+^ concentration with an increase in the flow rate and pause duration of PEF. This increase in concentration with an increase in flow rate can be associated with more intensive solution mixing, which facilitates the ions’ supply from the bulk solution to the CEM surface. The effect of the long pause duration of PEF on the increase in Na^+^ concentration variation is similar to an effect of flow rate increasing and consists in the fact that during pauses when the current is equal to zero, the stream continues to circulate and supplies fresh solution to the membrane.

The CEM membrane is not permeable for the Cl^–^ while P is fixed on the phosphoseryl residues, so phosphorus cannot migrate through the CEM. This explains that there is no significant difference (*p* > 0.05) in variations of Cl^-^ and P concentrations regardless of the flow rate and PEF regime applied. Concerning K^+^ and Ca^2+^ ions, ANOVA analysis also showed that PEF regime, as well as flow rate, had no significant impact on the variations of concentration. Thus, based on the ion migration results it can be concluded that only sodium migration is significant depending on the flow rate and PEF regime applied during EDBM.

### 3.5. Membrane Parameters

Despite the fact that the tested CEMs were cut from different membrane sheets, as well as BPMs, their initial thicknesses were equal in the range of SD and amounted to 0.152 ± 0.003 mm and 0.242 ± 0.004 mm for CEMs and BPMs, respectively (Appendix A in Appendix A). Statistical analysis showed that the thickness of all the membranes tested did not change after ED treatment (*p* > 0.05) (Appendix A). The result that thicknesses of BPMs 1, which were in contact with the caseinate stream and were fouled on the surface, did not change after the EDBM treatment can be connected with the fact that the sediments were removed from the BPM surface before thickness measurements in order to have their precise weight. In addition, since the thickness did not vary after removal of the fouling layer, it can be concluded that the fouling was mainly on the surface of the membrane and not inside the pores. Indeed, an inside fouling of the membrane would have affected the thickness, probably by swelling, even after collection of fouling from the membrane surface.

As well as for membrane thicknesses, the initial conductivities of CEMs and BPMs were identical in the range of SD, regardless of the batch, and amounted to 8.85 ± 0.27 mS/cm and 5.05 ± 0.51 mS/cm for CEMs and BPMs, respectively (Appendix A). Concerning the conductivity of CEM, ANOVA showed a decrease in membrane conductivity (Appendix A) after all the experimental conditions were considered (*p* < 0.001). It also appeared from the ANOVA results that there is no significant impact of the flow rate (*p* = 0.082), but PEF regimes (*p* = 0.007) have a significant impact on the membrane conductivity. The decrease in CEM conductivity was probably due to the substitution of relatively mobile Na^+^ by Ca^2+^ from the caseinate stream. The same effect of sodium concentration decrease and calcium concentration increase was described in [37] for the CMX membrane after EDBM of skim milk.

Statistical analysis (Appendix A) also showed that the conductivity of BPM 1 and BPM 2 increases after ED treatment (*p* < 0.001). As well, ANOVA analysis showed that the flow rate is a significant factor (*p* = 0.015) in terms of BPM 1 conductivity. However, the difference in conductivity is in the range of SD, hence, it can be concluded that the conductivity of BPM 1, as well as BPM 2, did not change after the experiment. As already observed by Ruiz and coauthors [19] the amount of fouling does not affect the conductivity of the membranes during ED process of a caseinate solution, regardless of the PEF mode applied, but in case of AEM protein fouling.

### 3.6. Energy Consumptions

The calculation of the EC according to Equation (4) surprisingly shows that the choice of the PEF regime does not affect the EC value. For all the cases considered, the EC was 0.136 ± 0.002 Wh, regardless of the PEF pause duration. This is probably connected with the fact that the amount of fouling formed on the BPM surface in a relatively short time of the experiment was small. This amount was not sufficient to significantly change the resistance of the membrane stack in all studied regimes. However, the EC normalized by the mole number of Na^+^ ions transported through the CEM, which separates the caseinate compartment and KCl compartment, decreases with increasing the flow rate and pause duration (Figure 9). At low flow rates and short PEF pause durations, the limiting current density across the CEM, *i*_lim_, is relatively low and the applied current density (5 mA/cm^2^) is apparently significantly higher than *i*_lim_. The transfer of ions through the CEM in these conditions occurs under a high concentration polarization. A high water splitting rate at the CEM reduces the sodium transport number in this membrane. With increasing flow rate and PEF pause duration, the *i*/*i*_lim_ ratio increases and the concentration polarization and water splitting rate decrease, so that the contribution of Na^+^ to the charge transfer through the CEM increases. The latter is clearly shown in Table 3. An increase in the amount of Na^+^ ions transported through the CEM in conditions of a constant summary EC results in decreasing the EC normalized by the Na^+^ transported mole number.

It should be mentioned that energy consumed for pumping a fluid increases with increase in Reynolds number in the ED setup. However, its contribution is generally insignificant, especially in the cases where BPMs are used in the stack, since the potential drop across these membranes is higher than across the monopolar membranes [38].

## 4. Conclusions

In the current study, it was demonstrated that the hydrodynamic conditions and parameters of PEF current mode have a strong effect on the kinetics of protein fouling on the cationic layer of a BPM, which was in contact with the caseinate stream during EDBM. It was observed that an increase in the conductivity of the caseinate stream was mainly influenced by water dissociation in BPM 1. Moreover, H^+^ generation also leads to the acidification of caseinate stream and protein fouling formation on the membrane surface. The latter occurs due to the interaction of H^+^ ions and caseinate anions, when locally (at the BPM cationic layer surface) their concentrations exceed a critical value. In addition, there is a process of caseinate desalination during EDBM, the mineral composition results confirmed that fact by demonstrating an increase in Na^+^ concentration in KCl with a significant effect of flow rate and PEF regime during EDBM. Ion migration through the CEM from the desalination to concentration chamber contributed to an increase in the KCl stream conductivity. Besides, OH^-^ electrogeneration from water splitting in BPM 2 leads to the basification of KCl stream and had an additional effect by contributing to the increase in the KCl stream conductivity.

Based on the obtained results, it can be concluded that increasing the pause duration and caseinate solution flow rate had a positive impact on the minimization of protein fouling on the cationic surface of BPM during EDBM of the model caseinate solution. Both a long pause and high flow rate contribute to a more effective decrease in the concentration of protons and caseinate anions at the BPM 1 surface. It was found that a very good result regarding the decrease in the amount of protein deposit was achieved with 50 s of pause duration of PEF and a flow rate corresponding to Re = 374. A further increase in flow rate in the ED system did not lead to a noticeable decrease in the amount of fouling. However, it should be mentioned that no conditions to completely eliminate protein fouling were found.

The next step in the work will be mathematical modelling of the process under study, which will be carried out in order to better understand the mechanism of fouling attachment and growth during EDBM.

## Figures and Tables

**Figure 1 membranes-11-00534-f001:**
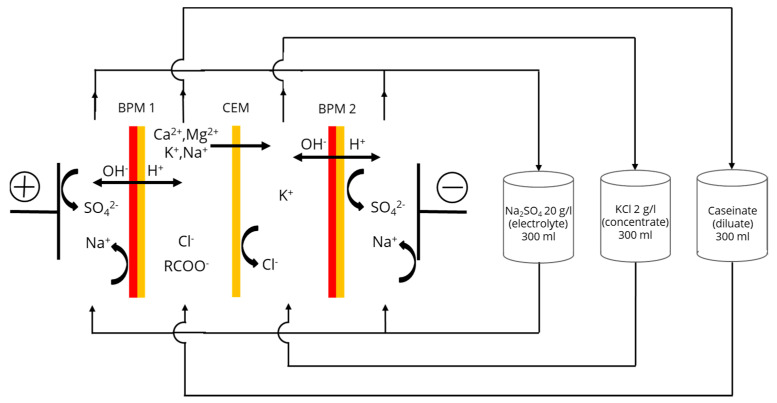
ED cell configuration for caseinate solution electroacidification by EDBM process.

**Figure 2 membranes-11-00534-f002:**
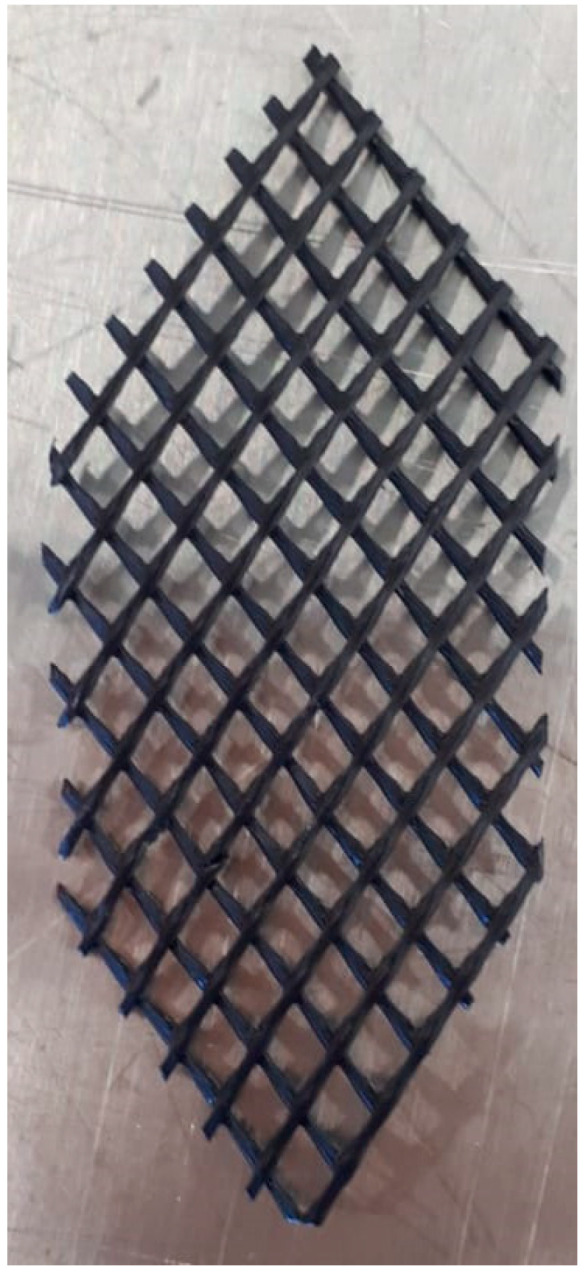
Photograph of the spacer used inside the ED channels.

**Figure 3 membranes-11-00534-f003:**
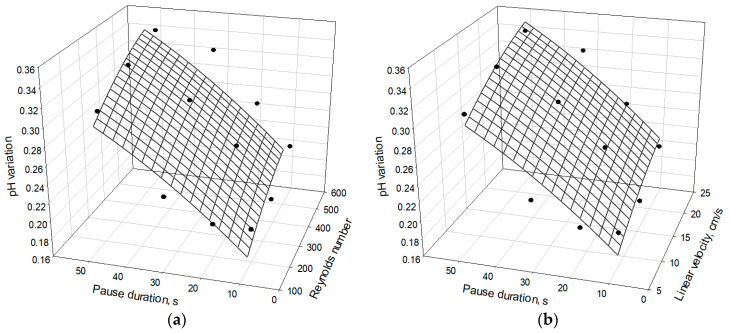
pH changes in the stream of caseinate solution during EDBM under PEF processes, each of which consumes 90 C of electricity, at different pause lapse durations (in s) and flow rates (expressed in: (**a**) Reynolds numbers and (**b**) linear velocity, cm/s). Reynolds numbers of 187, 374 and 560 correspond to linear velocities of 7.8, 15.6 and 23.4 cm/s, respectively. The pH value in the initial caseinate solution was 6.47 ± 0.02.

**Figure 4 membranes-11-00534-f004:**
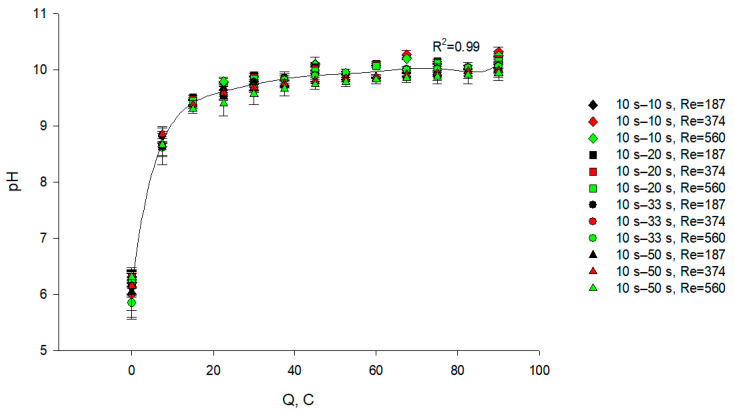
Evolution of pH in the KCl stream as a function of the amount of transferred charges (in C) during EDBM in PEF mode carried out at different flow rates (expressed in Reynolds numbers) and different pause durations; the regimes studied: 10 s–10 s, 10 s–20 s, 10 s–33 s and 10 s–50 s. Reynolds numbers of 187, 374 and 560 correspond to linear velocities of 7.8, 15.6 and 23.4 cm/s, respectively.

**Figure 5 membranes-11-00534-f005:**
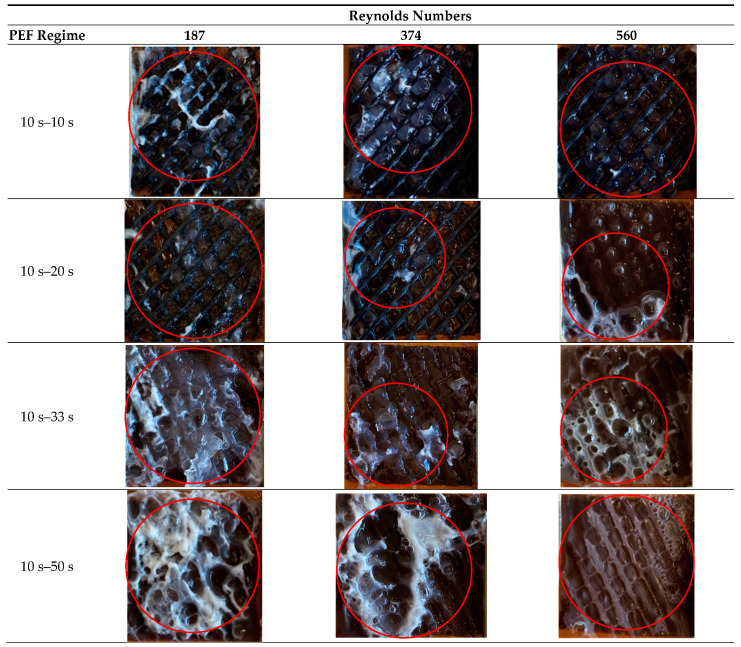
Photographs of the BPM cationic surface in contact with caseinate after EDBM in PEF mode carried out at different flow rates (corresponding to indicated Reynolds numbers) and different pause durations. Reynolds numbers of 187, 374 and 560 correspond to linear velocities of 7.8, 15.6 and 23.4 cm/s, respectively. The red circles indicate areas with the highest amount of deposit.

**Figure 6 membranes-11-00534-f006:**
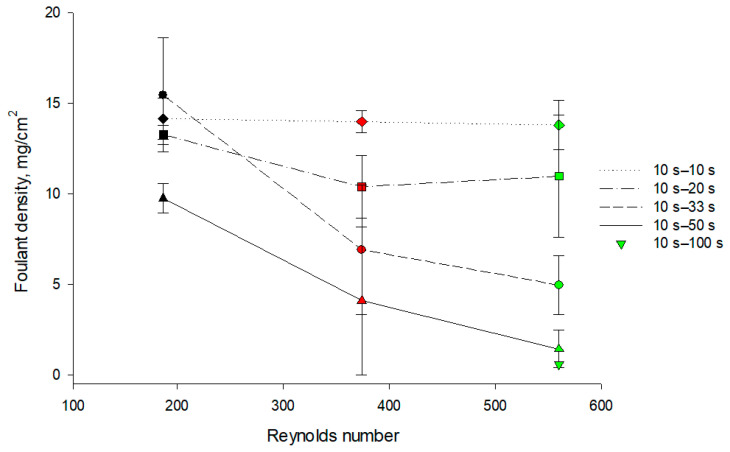
Weight of protein fouling normalized by the effective membrane area (in mg/cm^2^) recovered on the BPM cationic surface in contact with caseinate stream after EDBM depending on the Reynolds numbers in different PEF current modes with pulse–pause durations of 10 s–10 s, 10 s–20 s, 10 s–33 s, 10 s–50 s and 10 s–100 s. Reynolds numbers of 187, 374 and 560 correspond to linear velocities of 7.8, 15.6 and 23.4 cm/s, respectively.

**Figure 7 membranes-11-00534-f007:**
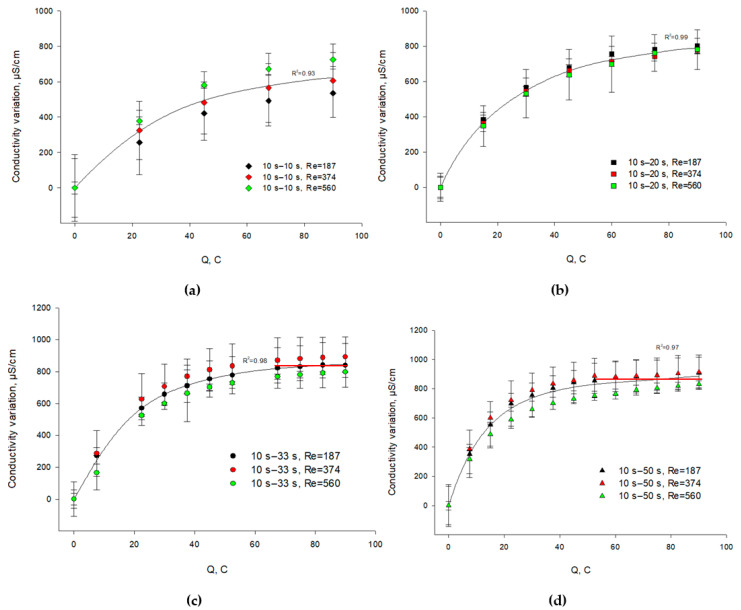
Evolution of caseinate stream conductivity variation as a function of the amount of transferred charges (in C) during EDBM in PEF mode carried out at different flow rates (expressed in Reynolds numbers) and different pause durations; the regimes studied: (**a**) 10 s–10 s, (**b**) 10 s–20 s, (**c**) 10 s–33 s and (**d**) 10 s–50 s. Reynolds numbers of 187, 374 and 560 correspond to linear velocities of 7.8, 15.6 and 23.4 cm/s, respectively. The red lines are added for the visualization of the steady-state plateau of the caseinate solution conductivity.

**Figure 8 membranes-11-00534-f008:**
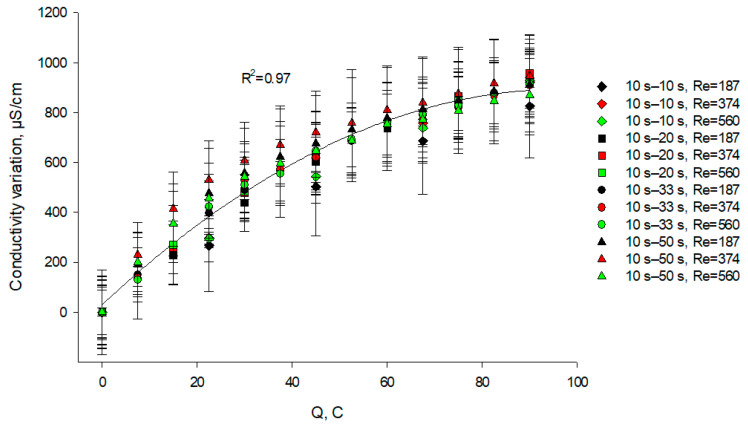
Evolution of KCl stream conductivity variation as a function of the amount of transferred charges (in C) during EDBM in PEF mode carried out at different flow rates (expressed in Reynolds numbers) and different pause durations; the regimes studied: 10 s–10 s, 10 s–20 s, 10 s–33 s and 10 s–50 s. Reynolds numbers of 187, 374 and 560 correspond to linear velocities of 7.8, 15.6 and 23.4 cm/s, respectively.

**Figure 9 membranes-11-00534-f009:**
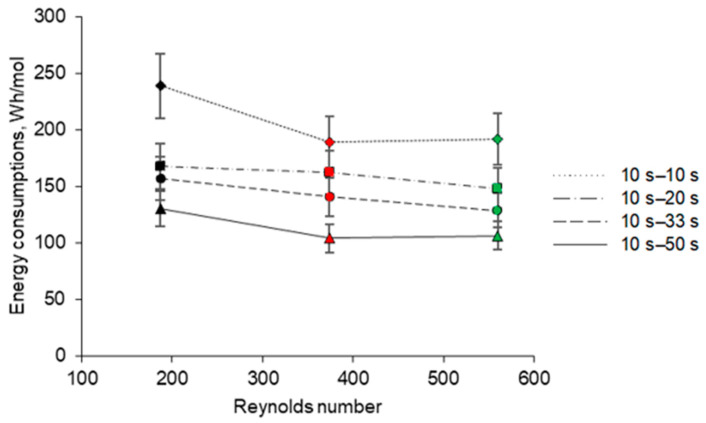
Energy consumptions normalized by the mole number of Na^+^ ions transported through the CEM (in Wh/mol) during EDBM depending on the Reynolds numbers under different PEF current modes with pulse–pause durations of 10 s–10 s, 10 s–20 s, 10 s–33 s and 10 s–50 s. Reynolds numbers of 187, 374 and 560 correspond to linear velocities of 7.8, 15.6 and 23.4 cm/s, respectively.

**Table 1 membranes-11-00534-t001:** Average mineral composition of raw caseinate solution.

Na, ppm	K, ppm	Mg, ppm	Ca, ppm	P, ppm
359.4	0.77	0.32	7.58	211.9

**Table 2 membranes-11-00534-t002:** Casein subunits characteristics.

Subunit	MW (kDa)	pI	Phosphates/Mole
α-s1	22–23.7	4.2–4.7	8–10
α-s2	25	-	10–13
Β	24	4.6–5.1	4–5
Κ	19	4.1–5.8	1

**Table 3 membranes-11-00534-t003:** Average mineral content of the KCl stream before and after EDBM depending on the Reynolds numbers and the pulse–pause durations of the PEF current modes (10 s–10 s, 10 s–20 s, 10 s–33 s and 10 s–50 s).

		Ca (ppm)			K (ppm)			Na (ppm)			P (ppm)			Cl (ppm)		
PEF Ratio	Re	Initial	Final	Variation	Initial	Final	Variation	Initial	Final	Variation	Initial	Final	Variation	Initial	Final	Variation
10 s–10 s	187	0.317 ± 0.090	0.115 ± 0.171	−0.202 ± 0.202 Aa	610 ± 6	596 ± 33	−13 ± 30 Aa	18.8 ± 6.2	62.1 ± 9.8	43.3 ± 5.3 Aa	0.023 ± 0.009	0.039 ± 0.013	0.016 ± 0.020 Aa	669 ± 21	653 ± 55	−16 ± 47 Aa
	374	0.436 ± 0.141	0.064 ± 0.070	−0.372 ± 0.076 Aa	670 ± 18	626 ± 35	−44 ± 34 Aa	23.3 ± 3.0	78.0 ± 3.6	54.8 ± 1.9 Aa	0.035 ± 0.003	0.058 ± 0.010	0.018 ± 0.006 Aa	727 ± 10	698 ± 51	−29 ± 40 Aa
	560	0.297 ± 0.047	0.025 ± 0.004	−0.272 ± 0.043 Aa	643 ± 31	620 ± 35	−23 ± 29 Aa	16.3 ± 3.1	71.7 ± 6.4	55.4 ± 3.8 Aa	0.055 ± 0.032	0.046 ± 0.036	−0.019 ± 0.0031 Aa	694 ± 14	704 ± 58	10 ± 44 Aa
10 s–20 s	187	0.194 ± 0.049	0.021 ± 0.004	−0.173 ± 0.045 Aa	684 ± 47	742 ± 73	58 ± 40 Aa	16.4 ± 5.3	77.4 ± 2.4	61.0 ± 4.4 Ba	0.041 ± 0.011	0.056 ± 0.032	0.016 ± 0.043 Aa	689 ± 31	675 ± 20	−14 ± 29 Aa
	374	0.267 ± 0.019	0.029 ± 0.021	−0.238 ± 0.098 Aa	741 ± 59	691 ± 54	−50 ± 40 Aa	19.3 ± 3.7	84.2 ± 9.0	64.9 ± 5.2 Ba	0.025 ± 0.006	0.030 ± 0.011	0.010 ± 0.017 Aa	659 ± 42	733 ± 43	74 ± 72 Aa
	560	0.245 ± 0.025	0.029 ± 0.007	−0.216 ± 0.023 Aa	721 ± 74	728 ± 35	7 ± 46 Aa	15.7 ± 2.7	87.0 ± 3.2	71.3 ± 5.3 Ba	0.015 ± 0.001	0.048 ± 0.021	0.034 ± 0.030 Aa	642 ± 7	699 ± 59	57 ± 66 Aa
10 s–33 s	187	0.186 ± 0.063	0.015 ± 0.004	−0.170 ± 0.060 Aa	699 ± 50	691 ± 45	−8 ± 43 Aa	15.3 ± 1.9	81.2 ± 6.2	65.9 ± 4.3 Ca	0.016 ± 0.003	0.046 ± 0.007	0.029 ± 0.005 Aa	669 ± 40	666 ± 42	−4 ± 26 Aa
	374	0.293 ± 0.127	0.035 ± 0.022	−0.258 ± 0.110 Aa	776 ± 64	697 ± 22	−79 ± 46 Aa	17.2 ± 6.8	91.4 ± 6.0	74.2 ± 2.1 Cab	0.015 ± 0.001	0.043 ± 0.025	0.028 ± 0.025 Aa	665 ± 44	706 ± 27	41 ± 25 Aa
	560	0.216 ± 0.115	0.029 ± 0.022	−0.187 ± 0.094 Aa	729 ± 52	728 ± 53	−2 ± 35 Aa	17.1 ± 2.3	98.6 ± 9.8	81.4 ± 8.5 Cb	0.025 ± 0.011	0.066 ± 0.017	0.041 ± 0.020 Aa	669 ± 74	703 ± 39	33 ± 71 Aa
10 s–50 s	187	0.178 ± 0.040	0.022 ± 0.009	−0.156 ± 0.034 Aa	730 ± 48	689 ± 31	−42 ± 18 Aa	14.5 ± 2.4	95.0 ± 3.0	80.5 ± 3.5 Da	0.019 ± 0.004	0.029 ± 0.006	0.011 ± 0.004 Aa	631 ± 25	670 ± 115	39 ± 92 Aa
	374	0.240 ± 0.009	0.059 ± 0.014	−0.182 ± 0.013 Aa	770 ± 10	765 ± 64	−4 ± 61 Aa	14.5 ± 1.4	113.6 ± 11.2	99.1 ± 10.0 Dbc	0.025 ± 0.008	0.065 ± 0.022	0.040 ± 0.029 Aa	696 ± 100	713 ± 108	17 ± 15 Aa
	560	0.442 ± 0.214	0.078 ± 0.006	−0.364 ± 0.208 Aa	757 ± 31	716 ± 69	−41 ± 100 Aa	20.2 ± 3.1	116.5 ± 15.9	96.4 ± 12.9 Dc	0.017 ± 0.016	0.049 ± 0.016	0.032 ± 0.016 Aa	736 ± 73	749 ± 99	14 ± 26 Aa

* Uppercase letters (A, B, C, D) indicate significant differences in PEF regimes for variations of concentrations for each mineral, lowercase letters (a, b, c) indicate differences in flow rate within one PEF regime. Reynolds numbers of 187, 374 and 560 correspond to the linear velocities of 7.8, 15.6 and 23.4 cm/s, respectively. The concentrations of Ca^2+^, K^+^, Na^+^ and P were obtained by ICP-EOS; Cl^-^ concentration determined by flow injection analysis.

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
