# Peer review of "Fouling Mitigation by Optimizing Flow Rate and Pulsed Electric Field during Bipolar Membrane Electroacidification of Caseinate Solution"

_membranes, 2021, doi:10.3390/membranes11070534_

Round 1

Reviewer 1 Report

In the manuscript, the authors proposed a strategy of high flow rate and long pause pulsed electric field to mitigate the protein fouling during bipolar membrane electrodialysis. It is found that increasing the pause duration and caseinate solution flow rate had a positive impact on the minimization of protein fouling occurring on the cationic surface of the bipolar membrane (BPM) during the EDBM. Both a long pause and high flow rate contribute to a more effective decrease in the concentration of protons and caseinate anions at the BPM surface. Also, the experimental conditions were optimized systematically. This work is very interesting, which can give a guidance to solve the membrane fouling in electrodialysis process. So, I recommend the publication of this work. However, some issues should be addressed before publication.

  1. Lines 51-53: Please cite relevant references for the sentence of “by increasing the diffusion permeability of membranes, decreasing their permselectivity, electrical conductivity and exchange capacity as well as increasing the energy costs of the whole ED process.”.
  2. Some minor mistakes about superscript in Lines 95, 449, 458, 466, should be revised.
  3. Line 121: It is better to give the schematic diagram of EDBM setup (microflow-type cell).
  4. Lines 149 and 150: Please give the size and photograph of the used spaces. In this manuscript, the linear velocity is so high (7.8-23.5 cm/s), does the solution flow evenly through the spaces?
  5. Line 156: What is the difference between conventional ED and EDBM?
  6. Line 217: Energy consumption calculation should be revised in the unit of Wh/mol or Wh/kg rather than Wh.
  7. Line 254: The equation number should be revised as 6.
  8. Line 313: In figure 4, the picture in the conditions of 10s-20s/162 is the same as the picture in the conditions of 10s-33s/162. Please the authors check the original photographs.
  9. Lines 353-358: The data in Figure 5 were not clear, especially for 10s-100s, there is only a point at the Reynolds number of 485. Please check the experiment data carefully.
  10. Lines 476 and 477: The sentences of “Reynolds numbers of 162, 323 and 485 correspond to the linear velocities of 7.8, 15.6 and 23.4 cm/s respectively. The concentrations of Ca2+, K+, Na+ and P were obtained by ICP-EOS; Cl- concentration determined by flow injection analysis.” can be removed to the note of Table 3.
  11. In section 3.6, more pumping energy are needed as the Reynolds number increases, so I suggest the pumping energy should be considered in the discussion of energy consumption.

Reviewer 2 Report

The manuscript was well written and data are clearly presented and discussed. Several comments are listed below:

  • Title: it suggests changing to “Fouling Mitigation by Optimizing Flow Rate and Pulsed Electric Field during Bipolar Membrane Electroacidification of Caseinate Solution”.
  • Abstract: providing the full name of the abbreviation.
  • Line 28, “the best membrane performance”.
  • Line 42-44, it is a repeated sentence to that in Line 40-41.
  • Line 100, “which was described in a previous study”.
  • The description of the terms in Eq 1 shall be provided.
  • Line 200, “foulant amount”
  • Line 209, “foulant deposition”
  • Line 211, it is not correct title.
  • As both density and viscosity are parameters relating to temperature, the authors may calculate the averaged viscosity due to varying temperature in this study. It may more meaningful to use velocity to discuss instead of Re Number.
  • Figure XX shall be referred in the text when it was discussed.
  • Line 247, the sentence needs to be rephrased.
  • It suggests presenting velocity instead of Re number in Figure 2 as the discussion in the text focused on the velocity.
  • Title of 3.2, “Cake layer foulant morphology and density”
  • Figure 5, “foulant density” in y-axis.
  • It suggests dividing Figure 6 and 7 into four sub-figures respectively in order to clearly present the data.
  • The conclusions shall be shortened to highlight the main findings.

Reviewer 3 Report

The work "Application of High Flow Rate and Long Pause Pulsed Electric Field Drastically Mitigates Protein Fouling Formation during Bipolar Membrane Electroacidification of a Model Caseinate Solution" has an interesting and an actual subject of the membrane research field.

  • The general presentation, structure and work organization were very well.
  • The methodology, apparatus and procedures wer adequate.
  • The results are interesting and useful.
  • The conclusions are based on obtained results.
  • Reference cover the specific field.

Author Response

Dear Reviewer,

Thank you very much for your time and consideration regarding our manuscript.

Best regards,

Authors